# Neutral bots probe political bias on social media

Wen Chen[1], Diogo Pacheco [1,2], Kai-Cheng Yang [1] & Filippo Menczer [1✉]

Social media platforms attempting to curb abuse and misinformation have been accused of political bias. We deploy neutral social bots who start following different news sources on Twitter, and track them to probe distinct biases emerging from platform mechanisms versus user interactions. We find no strong or consistent evidence of political bias in the news feed. Despite this, the news and information to which U.S. Twitter users are exposed depend strongly on the political leaning of their early connections. The interactions of conservative accounts are skewed toward the right, whereas liberal accounts are exposed to moderate content shifting their experience toward the political center. Partisan accounts, especially conservative ones, tend to receive more followers and follow more automated accounts. Conservative accounts also find themselves in denser communities and are exposed to more low-credibility content.

[1] Observatory on Social Media, Indiana University, Bloomington, IN, USA. [2] Department of Computer Science, University of Exeter, Exeter, UK.
✉email: fil@iu.edu

Compared with traditional media, online social media can connect more people in a cheaper and faster way than ever before. As a large portion of the population frequently use social media to generate content, consume information, and interact with others[1], online platforms are also shaping the norms and behaviors of their users. Experiments show that simply altering the messages appearing on social feeds can affect the online expressions and real-world actions of users[2,3] and that social media users are sensitive to early social influence[4,5]. At the same time, discussions on social media tend to be polarized around critical yet controversial topics like elections[6–8], vaccination[9], and climate change[10]. Polarization is often accompanied by the segregation of users with incongruent views into so-called echo chambers[11–16], homogeneous online communities that have been associated with ideology radicalization and misinformation spreading[17–20].

Countering such undesirable phenomena requires a deep understanding of their underlying mechanisms. On the one hand, online vulnerabilities have been associated with several socio-cognitive biases of humans[21–23], including the selection of belief-consistent information[24] and the tendency to seek homophily in social ties[25]. On the other hand, web platforms have their own algorithmic biases[26–28]. For example, ranking algorithms favor popular and engaging content, which may create a vicious cycle amplifying noise over quality[29]. Exposure to engagement metrics may also increase vulnerability to misinformation[30]. For a more extreme illustration, recent studies and media reports suggest that the YouTube recommendation system might lead to videos with more misinformation or extreme viewpoints regardless of the starting point[31].

Beyond the socio-cognitive biases of individual users and algorithmic biases of technology platforms, we have a very limited understanding of how collective interactions mediated by social media may bias the view of the world that we obtain through the online information ecosystem. The major obstacle is the complexity of the system — not only do users exchange huge amounts of information with large numbers of others via many hidden mechanisms, but these interactions can be manipulated overtly and covertly by legitimate influencers as well as inauthentic, adversarial actors who are motivated to influence opinions or radicalize behaviors[32]. Evidence suggests that malicious entities like social bots and trolls have already been deployed to spread misinformation and influence public opinion on critical matters[33–37].

In this study, we aim to reveal biases in the news and information to which people are exposed in social media ecosystems. We are particularly interested in clarifying the role of social media interactions during the polarization process and the formation of echo chambers. We therefore focus on U.S. political discourse on Twitter since this platform plays an important role in American politics, and strong polarization and echo chambers have been observed[6,38]. Twitter forms a directed social network where an edge from a friend node to a follower node indicates that content posted by the friend appears on the news feed of the follower.

Our goal is to study ecosystem bias, which includes both potential platform bias and the net effects of interactions with users of the social network (organic or not) that are mediated by the platform's mechanisms and regulated by its policies. While we only attempt to separate platform effects from naturally occurring biases in the narrow case of the feed curation, our investigation targets the overall bias experienced by users of the platform. This requires the exclusion of biases from individual users, which is a challenge when using traditional observational methods — it would be impossible to separate ecosystem effects from confounding factors that might affect the actions of tracked human accounts, such as age, gender, race, ideology, and socioeconomic status. We thus turn to a method that removes the need to control for such confounding factors by leveraging social media accounts that mimic human users but are completely controlled by algorithms, known as social bots[39]. Here we deploy social bots with neutral (unbiased) and random behavior as instruments to probe exposure biases in social media. We call our bots "drifters" to distinguish their neutral behavior from other types of benign and malicious social bots on Twitter[40].

The drifters are designed with an identical behavior model but with the only distinctive difference of their initial friend — the very first account they follow. After this initial action that represents the single independent variable (treatment) in our experiment, each drifter is let loose in the wild. To be sure, while all drifters have identical behaviors, their actions are different and depend on their initial conditions. We expect that a drifter who starts following liberal accounts will be more likely to be exposed to liberal content, to share some of this content, to be followed by liberal accounts, and so on. But these actions are driven by platform mechanisms and social interactions, not by political bias in the treatment: the behavioral model has no way of distinguishing between liberal, conservative, or any type of content. The drifter actions are therefore part of the dependent variables (outcomes) measured by our experiment.

This methodology allows us to examine the combined biases that stem both from Twitter's system design and algorithms, and from the organic and inorganic social interactions between the drifters and other accounts. Our research questions are: (i) How are influence and exposure to inauthentic accounts, political echo chambers, and misinformation impacted by early actions on a social media platform? And (ii) Can such differences be attributed to political bias in the platform's news feed?

To answer these questions, we initialized drifters from news sources across the political spectrum. After five months, we examined the content consumed and generated by the drifters and analyzed (i) characteristics of their friends and followers, including their liberal-conservative political alignment inferred by shared links and hashtags; (ii) automated activity measured via machine learning methods; and (iii) exposure to information from low-credibility sources identified by news and fact-checking organizations.

We find that the political alignment of the initial friend has a major impact on the popularity, social network structure, exposure to bots and low-credibility sources, and political alignment manifested in the actions of each drifter. However, we find no evidence that these outcomes can be attributed to platform bias. The insights provided by our study into the political currents of the Twitter's information ecosystem can aid the public debate about how social media platforms shape people's exposure to political information.

## Results

All drifters in our experiment follow the same behavior model, whose design is intended to be neutral, not necessarily realistic. Each drifter is activated at random times to performs actions. Action types, such as tweets, likes, and replies are selected at random according to predefined probabilities. For each action, the model specifies how to select a random target, such as a tweet to be retweeted or a friend to be unfollowed. Time intervals between actions are drawn from a broad distribution to produce realistic bursty behaviors. See "Methods" for further details.

We developed 15 drifter bots, divided them into five groups, and initialized each drifter in the same group with the same initial friend. Each Twitter account used as a first friend a popular news source aligned with the Left, Center-Left, Center, Center-Right, or Right of the U.S. political spectrum (see details in "Methods").

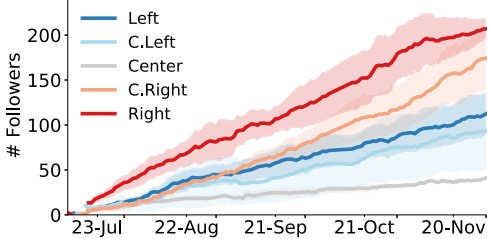

**Fig. 1 Growth in followers.** The *x*-axis displays the duration of the experiment in 2019 and the *y*-axis reports the average numbers of followers of different drifter groups. Colored confidence intervals indicate ±1 standard error. Source data are provided as a Source data file.

We refer to the drifters by the political alignment of their initial friends; for example, bots initialized with Center-Left sources are called "C. Left" drifters.

Between their deployment on July 10, 2019, and until their deactivation on December 1, 2019, we monitored the behaviors of the drifters and collected data on a daily basis. In particular, we measured: (1) the number of followers of each drifter to compare their ability to gain influence; (2) the echo-chamber exposure of each drifter; (3) the likely automated activities of friends and followers of the drifters; (4) the proportion of low-credibility information to which the drifters are exposed; and (5) the political alignment of the content generated by the drifters and their friends to probe political biases.

**Influence**. The number of followers can be used as a crude proxy for influence[41]. To gauge how political alignment affects influence dynamics, Fig. 1 plots the average number of followers of drifters in different groups over time. To compare the growth rates of different groups, we considered consecutive observations of the follower counts of each drifter and aggregated them across each group ($n = 387$ for Left, 373 for C. Left, 389 for Center, 387 for C. Right, and 386 for Right). Two trends emerged from *t*-tests (all *t*-tests in this and the following analyses are two-sided). First, drifters with the most partisan sources as initial friends tend to attract significantly more followers than Center drifters (d. f. = 774, $t = 5.13$, $p < 0.001$ for Left vs. Center and d.f. = 773, $t = 8.00$, $p < 0.001$ for Right vs. Center). Second, drifters with Right-leaning initial sources gain followers at a significantly higher rate than those with Left-leaning initial sources (d.f = 771, $t = 3.84$, $p < 0.001$ for Right vs. Left). More details and robustness analysis of these findings are presented in Supplementary Notes.

The differences in influence among drifters could be affected not only by the political alignment but also by other characteristics of their initial friends. To disentangle these factors, we measured the correlation between the number of drifter followers and two features of their initial friends: their overall influence and their popularity among other politically aligned accounts. While drifter influence is not affected by the overall influence of the initial friends, it is positively correlated with their popularity among politically aligned accounts (see Supplementary Notes). This is consistent with evidence that users with shared partisanship are more likely to form social ties[42], as we explore next.

**Echo chambers**. We define echo chambers as dense and highly clustered social media neighborhoods that amplify exposure to homogeneous content. To investigate whether the drifter bots find themselves in such echo chambers, let us consider the ego network of each drifter, i.e., the network composed by the drifter and its friends and followers. We can use density and transitivity of ego networks as proxies for the presence of echo chambers.

Density is the fraction of node pairs that are connected in a network. Transitivity measures the fraction of possible triangles that are actually present among the nodes. High transitivity means that friends and followers are likely to follow each other too. See "Methods" for further details.

Figure 2a, b shows the average density and transitivity of ego networks for the drifters (see details in "Methods"). Since the two metrics are correlated in an ego network, Fig. 2c also plots the transitivity rescaled by that of shuffled random networks (see "Methods"). The ego networks of Right drifters are more dense than those of Center drifters (d.f. = 4, $t = -8.28$, $p = 0.001$), whereas the difference in density is not significant between Center and Left drifters (d.f. = 4, $t = -2.68$, $p = 0.055$). Right account networks also have higher transitivity than Center networks (d.f. = 4, $t = -9.31$, $p < 0.001$); so do the Left account networks (d.f. = 4, $t = -3.53$, $p = 0.024$). Right accounts are more clustered than centrists even when accounting for the difference in density (d.f. = 4, $t = -8.96$, $p < 0.001$) while the difference is not significant for Left vs. Center (d.f. = 4, $t = -2.73$, $p = 0.053$). Furthermore, Right drifters are in stronger echo chambers than Left drifters (d.f. = 4, $t = -3.84$, $p = 0.019$ for density and d.f. = 4, $t = -3.02$, $p = 0.039$ for transitivity). However, the difference in normalized transitivity between Left and Right is not significant (d.f. = 4, $t = -0.60$, $p = 0.579$), indicating that the higher clustering on the Right is explained by the density of social connections.

To get a better sense of what these echo chambers look like, Fig. 2d maps the ego networks of the 15 drifters. In addition to the clustered structure, we observe a degree of homogeneity in shared content as illustrated by the colors of the nodes, which represent the political alignment of the links shared by the corresponding accounts (see "Methods"; similar results are obtained by measuring political alignment based on shared hashtags). In general, the neighbors of a drifter tend to share links to sources that are politically aligned with the drifter's first friend. We note a few exceptions, however. The Left drifters and their neighbors are more moderate, having shifted their alignment toward the Center. One of the C. Left drifters has become connected to many conservative accounts, shifting its alignment to the Right. And one of the C. Right drifters has shifted its alignment to the Left, becoming connected to mostly liberal accounts after randomly following @CNN, a Left-leaning news organization. In most cases, drifters find themselves in structural echo chambers where they are exposed to content with homogeneous political alignment that mirrors their own.

**Automated activities**. Automated accounts known as social bots were actively involved in online discussions about recent U.S. elections[33,43,44]. It is therefore expected for the drifters to encounter bot accounts. We used the Botometer service[45,46] to collect the bot scores of friends and followers of the drifters. We report the distributions of bot scores for both friends and followers of the drifters in Fig. 3. Unsurprisingly, drifters are more likely to have bots among their followers than among their friends, across the political spectrum. Focusing on friends reveals a more serious potential vulnerability of social media users. We find that accounts followed by partisan drifters are more bot-like than those followed by centrist drifters (d.f. = 618, $t = -6.14$, $p < 0.001$ for Right vs. Center and d.f. = 486, $t = -3.67$, $p < 0.001$ for Left vs. Center). Comparing partisans and moderates, Right drifters follow accounts that are more bot-like than C. Right drifters (d.f. = 735, $t = -3.01$, $p = 0.003$), while the difference is smaller on the liberal side (d.f. = 541, $t = -2.56$, $p = 0.011$ for Left vs. C. Left). Among partisans, Right drifters follow accounts that are slightly more bot-like than Left ones (d.f. = 694, $t = -2.33$, $p = 0.020$).

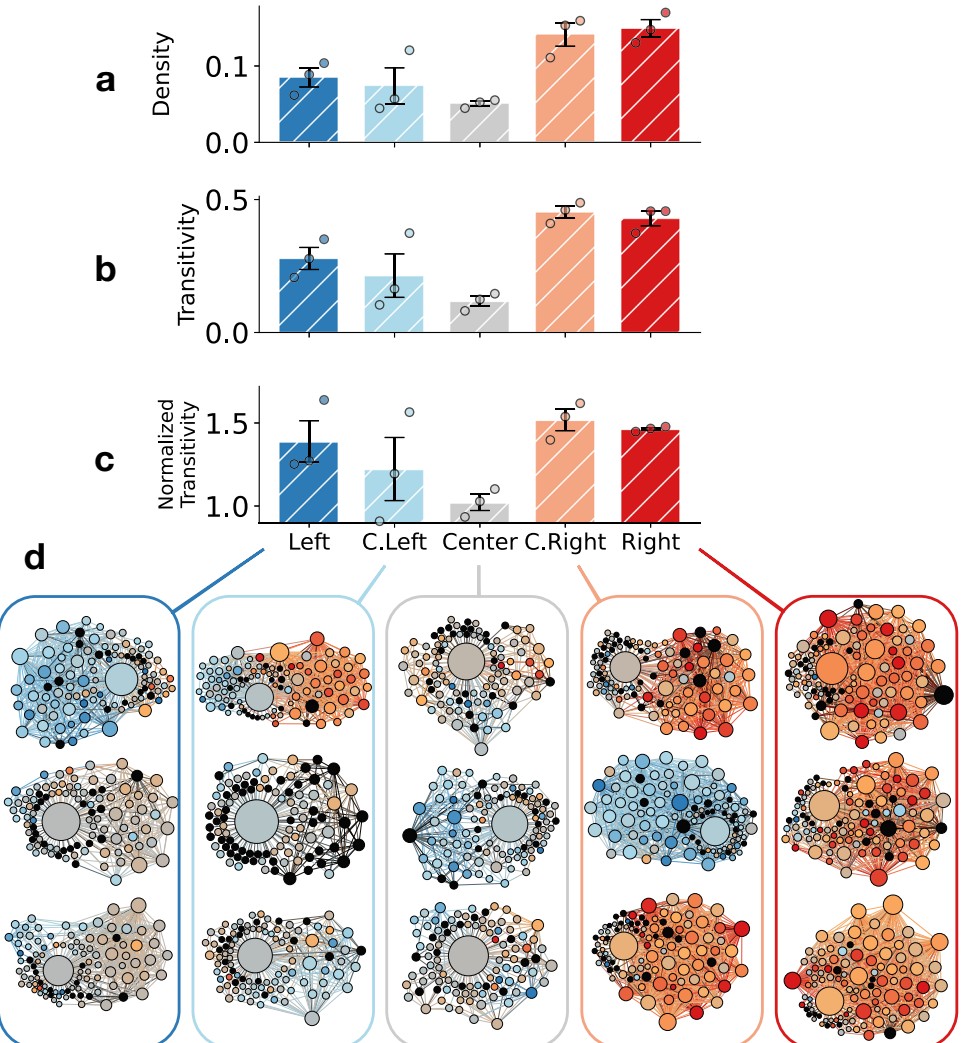

**Fig. 2 Echo chamber structure around drifters. a** Density, **b** transitivity, and **c** normalized transitivity of drifters ego networks in different groups. Error bars indicate the standard errors ($n = 3$ drifters in each group). **d** Ego networks of the drifters in the five groups. Nodes represent accounts and edges represent friend/follower relations. Node size and color represent degree (number of neighbors) and political alignment of shared links, respectively. Black nodes have missing alignment scores due to not sharing political content. Source data are provided as a Source data file.

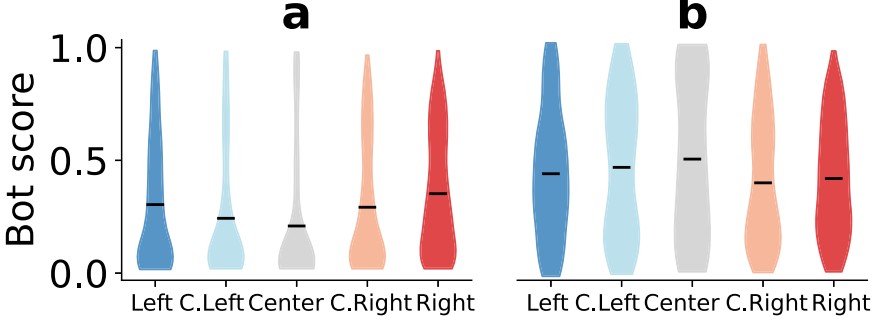

**Fig. 3 Distributions of the bot scores of friends and followers of drifters.** The bot score is a number between zero and one, with higher scores signaling likely automation. For each group, we consider the union of **a** friends and **b** followers of the drifters in that group. Bars indicate averages. For friends, $n = 282$ (Left), 261 (C. Left), 206 (Center), 323 (C. Right), and 414 (Right). For followers, $n = 172$ (Left), 118 (C. Left), 65 (Center), 205 (C. Right), and 299 (Right). Source data are provided as a Source data file.

**Exposure to low-credibility content**. Since the 2016 U.S. presidential election, concern has been heightened about the spread of misinformation in social media[47]. We consider a list of low-credibility sources that are known to publish false and misleading news reports, conspiracy theories, junk science, and other types of misinformation (details about low-credibility sources are found in "Methods"). We analyze exposure to content from these low-credibility sources for different groups of drifters in Fig. 4. We observe that Right drifters receive more low-credibility content in their news feeds than other groups ($t = 5.06, p = 0.007$ compared

to C. Right; $p < 0.001$ for other groups: $t = 27.47$ compared to Center, $t = 15.06$ to C. Left, and $t = 13.14$ to Left; d.f = 4 in all cases). Almost 15% of the links that appear in the timelines of Right drifters are from low-credibility sources. We also measured the absolute number of low-credibility links, and used the total number of tweets or the number of tweets with links as the denominator of the proportions; the same pattern emerges in all cases.

We used @BreitbartNews as the initial friend account for Right drifters because it is one of the most popular conservative news sources. Although *Breitbart News* appears in lists of hyper-partisan sources used in the literature, to prevent biasing our

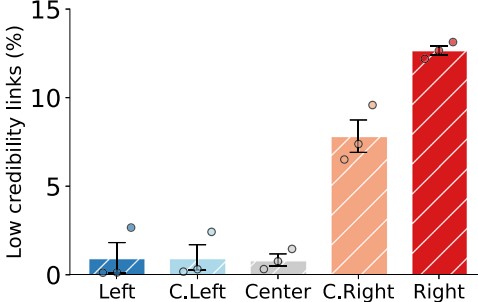

**Fig. 4 Exposure to low-credibility content.** The bars express the proportions of low-credibility links in the home timelines of different drifter groups. Error bars indicate standard errors ($n = 3$ drifters in each group). Source data are provided as a Source data file.

results, *Breitbart News* is not labeled as a low-credibility source in this analysis and does not contribute to the proportions in Fig. 4.

**Political alignment and news feed bias.** We wish to measure the political alignment of content consumed and produced by drifters. Given a link (URL), we can extract the source (website) domain name and obtain an alignment score based on its known political bias. Similarly, given a hashtag, we can calculate a score based on a computational technique that captures the political alignment of different hashtags. These scores can then be averaged across the links or hashtags contained in a feed of tweets to measure their aggregate political alignment. Further details can be found in the Methods.

The *home timeline* (also known as news feed) is the set of tweets to which accounts are exposed. The *user timeline* is the set of tweets produced by an account. In Fig. 5a,b,d,e we observe how the political alignment of information to which drifters are exposed in their home timelines ($s_h$) and of the content generated by them in their user timelines ($s_u$) changed during the experiment. The initial friends strongly affect the political trajectories of the drifters. Both in terms of information to which they are exposed and content they produce, drifters initialized with Right-leaning sources stay on the conservative side of the political spectrum. Those initialized with Left-leaning sources, on the other hand, tend to drift toward the political center: they are exposed to more conservative content and even start spreading it. These findings are robust with respect to the method used to calculate political alignment, whether based on hashtags (Fig. 5a, b) or links (Fig. 5d, e).

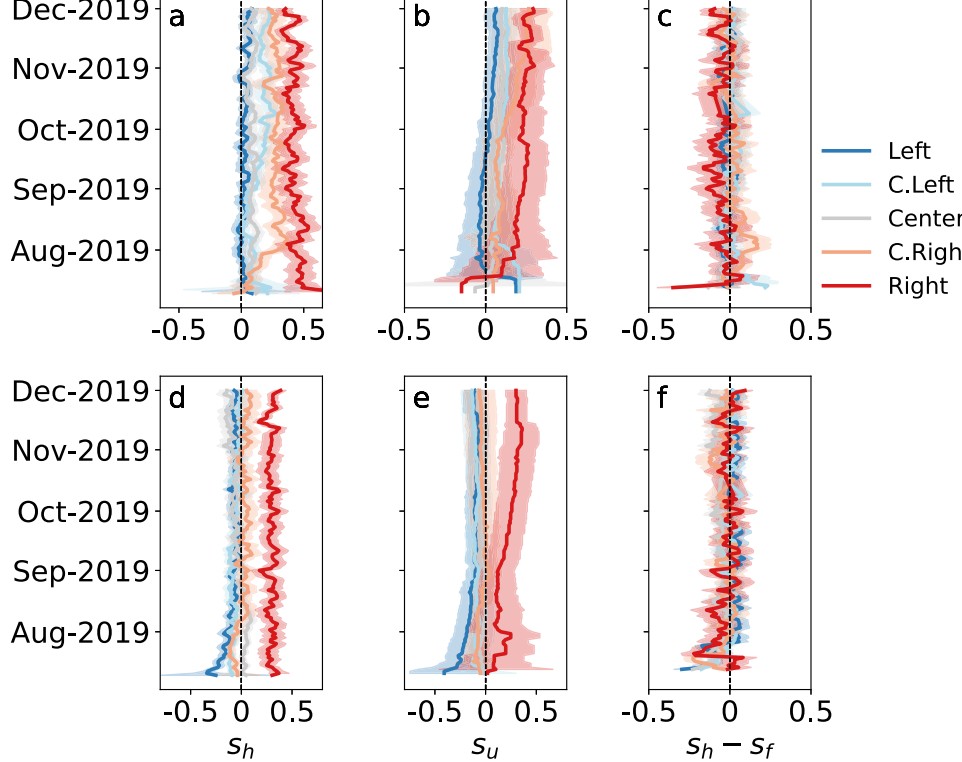

**Fig. 5 Time series of political alignment.** Negative alignment scores mean Left-leaning and positive scores mean Right-leaning. Alignment scores $s_h$ are calculated from content to which drifters are exposed in their *home* timelines, based on hashtags (**a**) and links (**d**). Alignment scores $s_u$ are calculated from content expressed by the drifter posts in their *user* timelines, based on hashtags (**b**) and links (**e**). The difference $s_h - s_f$ measures news feed bias experienced by drifters, where the political alignment score $s_f$ is derived from the content generated by *friends*, based on hashtags (**c**) and links (**f**). Missing values are replaced by preceding available ones. Colored confidence intervals indicate ± 1 standard error. Source data are provided as a Source data file. Plots for each individual drifter are in Supplementary Figs. 2–4.

We measured the political bias of the news feed by calculating the difference between the alignment score of tweets posted by friends of the drifters, $s_f$, and the score of tweets in the home timeline, $s_h$. The results are shown in Fig. 5c, f for alignment computed from hashtags and links, respectively. In the case of hashtags, we observe little evidence of political bias by the news feed. For Right-leaning drifters, there is a small shift toward the Center, suggesting a weak bias of the news feed (Fig. 5c). To confirm this visual observation, we performed a paired $t$-test comparing the daily averages of home timeline scores $s_h$ and friend user timeline scores $s_f$ (Supplementary Table 3). The effect is small for all but the Right group of drifters ($p < 0.001$, Cohen's $d = 0.56$). Similarly, in the case of links (Fig. 5f), we observe Left bias for drifters in the Center group ($p < 0.001$, Cohen's $d = 0.76$). For the other groups, the effect is small (Supplementary Table 3).

Further details on the bias analysis can be found in Supplementary Notes, together with trajectories of individual drifters, data on follow-back rates, and descriptive statistics of the drifters.

## Discussion

The present results suggest that early choices about which sources to follow impact the experiences of social media users. This is consistent with previous studies[4,5]. But beyond those initial actions, drifter behaviors are designed to be neutral with respect to partisan content and users. Therefore the partisan-dependent differences in their experiences and actions can be attributed to their interactions with users and information mediated by the social media platform — they reflect biases of the online information ecosystem.

Drifters with Right-wing initial friends are gradually embedded into dense and homogeneous networks where they are constantly exposed to Right-leaning content. They even start to spread Right-leaning content themselves. Such online feedback loops reinforcing group identity may lead to radicalization[17], especially in conjunction with social and cognitive biases like in-/out-group bias and group polarization. The social network communities of the other drifters are less dense and partisan.

We selected popular news sources across the political spectrum as initial friends of the drifters. There are several possible confounding factors stemming from our choice of these accounts: their influence as measured by the number of followers, their popularity among users with a similar ideology, their activity in terms of tweets, and so on. For example, @FoxNews was popular but inactive at the time of the experiment. Furthermore, these quantities vary greatly both within and across ideological groups (Supplementary "Methods"). While it is impossible to control for all of these factors with a limited number of drifters, we checked for a few possible confounding factors. We did not find a significant correlation between initial friend influence or popularity measures and drifter ego network transitivity. We also found that the influence of an initial friend is not correlated with the drifter influence. However, the popularity of an initial friend among sources with similar political bias is a confounding factor for drifter influence. Online influence is therefore affected by the echo-chamber characteristics of the social network, which are correlated with partisanship, especially on the political Right[20,48]. In summary, drifters following more partisan news sources receive more politically aligned followers, becoming embedded in denser echo chambers and gaining influence within those partisan communities.

The fact that Right-leaning drifters are exposed to considerably more low-credibility content than other groups is in line with previous findings that conservative users are more likely to engage with misinformation on social media[20,49]. Our experiment suggests that the ecosystem can lead completely unbiased agents to this condition, therefore, it is not necessary to impute the vulnerability to characteristics of individual social media users. Other mechanisms that may contribute to the exposure to low-credibility content observed for the drifters initialized with Right-leaning sources involve the actions of neighbor accounts (friends and followers) in the Right-leaning groups, including the inauthentic accounts that target these groups.

Although *Breitbart News* was not labeled as a low-credibility source in our analysis, our finding might still be biased in reflecting this source's low credibility in addition to its partisan nature. However, @BreitbartNews is one of the most popular conservative news sources on Twitter (Supplementary Table 1). While further experiments may corroborate our findings using alternative sources as initial friends, attempting to factor out the correlation between conservative leanings and vulnerability to misinformation[20,49] may yield a less-representative sample of politically active accounts.

While most drifters are embedded in clustered and homogeneous network communities, the echo chambers of conservative accounts grow especially dense and include a larger portion of politically active accounts. Social bots also seem to play an important role in the partisan social networks; the drifters, especially the Right-leaning ones, end up following a lot of them. Since bots also amplify the spread of low-credibility news[33], this may help explain the prevalent exposure of Right-leaning drifters to low-credibility sources. Drifters initialized with far-Left sources do gain more followers and follow more bots compared with the Center group. However, this occurs in a way that is less emphatic and vulnerable to low-credibility content compared to the Right and Center-Right groups. Nevertheless, our results are consistent with findings that partisanship on both sides of the political spectrum increases the vulnerability to manipulation by social bots[50].

Twitter has been accused of favoring liberal content and users. We examined the possible bias in Twitter's news feed, i.e., whether the content to which a user is exposed in the home timeline is selected in a way that amplifies or suppresses certain political content produced by friends. Our results suggest this is not the case: in general, the drifters receive content that is closely aligned with whatever their friends produce. A limitation of this analysis is that it is based on limited sets of recent tweets from drifter home timelines ("Methods"). The exact posts to which Twitter users are exposed in their news feeds might differ due to the recommendation algorithm, which is not available via Twitter's programmatic interface.

Despite the lack of evidence of political bias in the news feed, drifters that start with Left-leaning sources shift toward the Right during the course of the experiment, sharing and being exposed to more moderate content. Drifters that start with Right-leaning sources do not experience a similar exposure to moderate information and produce increasingly partisan content. These results are consistent with observations that Right-leaning bots do a better job at influencing users[51].

In summary, our experiment demonstrates that even if a platform has no partisan bias, the social networks and activities of its users may still create an environment in which unbiased agents end up in echo chambers with constant exposure to partisan, inauthentic, and misleading content. In addition, we observe a net bias whereby the drifters are drawn toward the political Right. On the conservative side, they tend to receive more followers, find themselves in denser communities, follow more automated accounts, and are exposed to more low-credibility content. Users have to make extra efforts to moderate the content they consume and the social ties they form in order to counter these currents, and create a healthy and balanced online experience.

Given the political neutrality of the news feed curation, we find no evidence for attributing the conservative bias of the information ecosystem to intentional interference by the platform. The bias can be explained by the use (and abuse) of the platform by its users, and possibly to unintended effects of the policies that govern this use: neutral algorithms do not necessarily yield neutral outcomes. For example, Twitter may remove or demote information from low-credibility sources and/or inauthentic accounts, or suspend accounts that violate its terms. To the extent that such content or users tend to be partisan, the net result would be a bias toward the Center. How to design mechanisms capable of mitigating emergent biases in online information ecosystems is a key question that remains open for debate.

**Broader considerations and study limitations**. The methodology based on neutral social bots can be applied to study a number of biases of social media platforms and their information ecosystems, in addition to political bias. Applications of our method could be used to study gender and racial bias, hate speech, and algorithmic bias. Bots could also be used for benign interventions, such as posting content from trustworthy sources in response to harmful health misinformation. However, when bots interact with humans without informed consent, even in the absence of deception, there are trade-offs between potential benefits and risks to the users. For example, our neutral bots might share misinformation or contribute to polarization. Therefore, the ethical implications of such applications must be evaluated carefully.

Our main finding — that the information Twitter users see in their news feed depends strongly on the political leaning of their early connections — has important societal implications. It may increase awareness among social media users about the implicit biases of their online connections and their own vulnerabilities to selective exposure of information, or worse — influence campaigns, manipulation, misinformation, and polarization.

The absence of strong or consistent evidence of political bias in the Twitter news feed could help inform the public debate about social media platform regulation, and claims that platforms censor political speech.

The media have reported extensively on links between online misinformation and harmful behaviors in domains like health and elections. Our data are limited to online behaviors and cannot gauge the impact of political bias on real-world actions. Our research also does not address platform policies and their enforcement, nor other types of algorithmic bias. In particular, our study is unable to evaluate the effects of Twitter's personalized ranking of news feed posts, friend recommendations, suspension of accounts, or ads. A final limitation of our experiment is in the small number of deployed neutral bots, motivated by the ethical considerations discussed above.

Further research questions remain open concerning political bias in the information ecosystems. How would the findings be affected by deploying a larger number of bots and starting from news sources with greater or smaller diversity in popularity, influence, activity, or political slant? What is the net effect of major changes in platform policy/enforcement, such as the takedown of misinformation "superspreaders"? Or the migration of radical users to other platforms? Can the findings on Twitter be generalized to platforms with perhaps different user demographics or more partisan user populations? Finally, our study is U.S.-centric. Similar questions could be explored in the political contexts of other countries, possibly yielding different conclusions.

## Methods

Here we provide details about the design of drifter bots, the computation of political alignment metrics, the identification of low-credibility sources, and the

characterization of echo chambers. All drifter activities are managed through Twitter's application programmatic interface (API).

**Drifter behavior model**. Drifter bots are the key instrument for this study. They are designed to mimic social media users so that the data collected from their actions and interactions reflect experiences on the platform. The drifters are intended to have a neutral behavior, even if it is not necessarily realistic. For example, they lack any ability to comprehend the content to which they are exposed or the users with whom they interact. All actions are controlled by a stochastic model which was consistent across treatments and unchanged during the experiment.

The Twitter API was employed to control drifter bot behavior in compliance with Twitter terms of use for academic research. Twitter does not prohibit bots as long as they do not engage in any prohibited behaviors like spam, deception, and other abuse. Drifters did not engage in any such behaviors. In particular, to avoid impersonating any human, the drifter accounts were named after fictional robots in the arts and literature; used robot images in the public domain for their profiles; and used random quotes as profile descriptions, also avoiding any political references that would bias the experiment. In addition, the drifters did not engage in direct messages nor use any advanced natural language generative models. If a human user started a conversation with any of our drifters, it would be obvious to them that those accounts were bots.

The experimental protocol was vetted and approved by the Indiana University ethics board. A waiver of informed consent was granted: the protocol did not require disclosure to Twitter users with whom the bots interacted. However, a large number of drifters could have a negative impact by spreading misinformation, reinforcing echo chambers, or amplifying malicious accounts. Therefore, we deployed 15 drifters as a trade-off between limiting potential harm and achieving statistically significant results.

Like many human behaviors, social media activity is bursty[52]. To reproduce this feature, we draw time intervals $\Delta t$ between two successive actions from a power-law distribution $P(\Delta t) \sim \Delta t^{-\alpha}$, with the exponent $\alpha = 0.9$ manually tuned to minimize the bot score obtained from the Botometer service. The distribution was cut off at a maximum sleep duration of 7 h between consecutive actions. Intervals were further scaled to obtain an average frequency of 20 – 30 actions per day — a typical activity level of normal users estimated from sampled active Twitter accounts. Moreover, the drifters were inactive between midnight and 7 a.m., a typical sleep cycle for social media users[53].

Every time a drifter is activated, it randomly selects an action and a source as illustrated in Fig. 6. Actions include tweets, retweets, likes, replies, etc. Sources include the home timeline, trends, friends, etc. Each action is selected with a predefined probability. Given the selected action, one of a set of possible sources is selected with a predefined conditional probability. A random object is then drawn from the source and the action is performed on it. For example, if the action is a retweet and the source is the home timeline, then a random tweet in the drifter's home timeline is retweeted. Non-English sources (users and tweets) are disregarded when they can be identified from metadata. Finally, the bot sleeps for a random interval until the next action. To avoid behaviors typical of spambots that violate Twitter's policies, the follow and unfollow actions have additional constraints regarding the ratio between friends and followers. The constraints are mutually exclusive, so that if one of these two actions fails due to a constraint not being satisfied, the other action can be performed. Details about actions, sources, their associated probabilities, and constraints can be found in Supplementary Methods.

The only difference among the drifters was the way their friend lists were initialized. This was our experiment's independent variable or treatment. We started from Twitter accounts associated with established and active news sources with different political alignments. While mapping the political spectrum to a one-dimensional scale is reductive, we obtain manageable experimental treatments by selecting five sources: *The Nation* (Left), *The Washington Post* (Center-Left), *USA Today* (Center), *The Wall Street Journal* (Center-Right), and *Breitbart News* (Right). These sources were selected because they are among the most popular on Twitter, as well as among the most followed by users aligned with different portions of the U.S. political spectrum (Supplementary Table 1). The choice of the five accounts, however, was not based on maximizing any single criterion such as popularity. The 15 drifters were divided into five groups so that each of three bots in the same group started by following the same source account. The friend list of each drifter was then expanded by following a random sample of five English-speaking friends of the first friend, and a random sample of five English-speaking followers of the first friend — 11 accounts in total.

**Political alignment metrics and news feed bias**. Given our goal of gauging political bias, we need to measure the political alignment of tweets and accounts within the liberal-conservative spectrum. This alignment is operationally defined by a score between $-1$ (liberal) and $+1$ (conservative). We adopt two independent approaches, one based on hashtags and one on links, to ensure the robustness of our results. Both approaches start with assigning political alignment scores to entities that may be present in tweets, namely hashtags and links. See Supplementary "Methods" for details about how these entities are extracted from tweets. The entity scores are then averaged at the tweet level to obtain alignment scores for the tweets, and further at the user level to measure the political alignment of users.

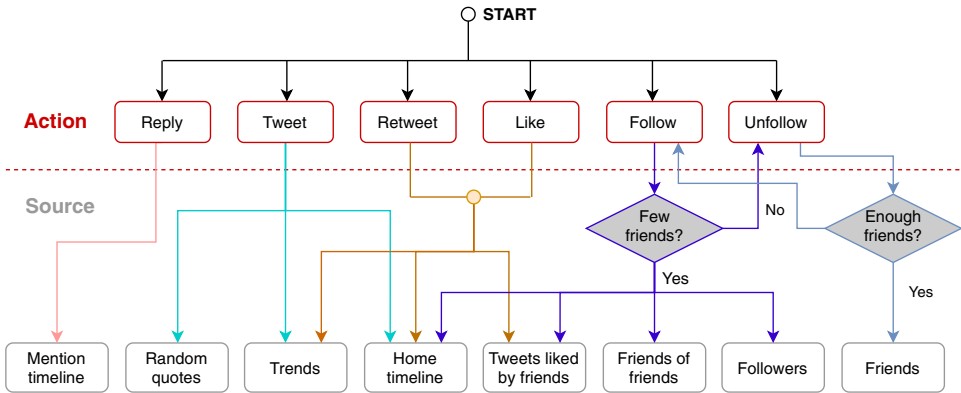

**Fig. 6 Drifter bot behavior model.** Each action box is connected with boxes that indicate the sources used for that action. For example, the source of a retweet can be a trending tweet, a tweet in the home timeline, or a tweet liked by a friend. Links to actions and sources are associated with the probabilities. Follow and unfollow actions require additional constraints to be satisfied (gray diamonds).

The hashtag-based approach relies on hashtags (keywords preceded by the hash mark #) commonly included by users in their social media posts because they are concise and efficient ways to label topics, ideas, or memes so that others can find the messages. Hashtags are often used to signal political identities, beliefs, campaigns, or alignment[54]. We apply the word2vec algorithm[55] to assign political alignment scores to hashtags in a semi-supervised fashion. word2vec maps words in the text to continuous vector representations, which have been shown to capture semantic relations between the words[56]. The axis between a pair of carefully selected word vectors can encode a meaningful cultural dimension and an arbitrary word's position along this axis reflects its association with this cultural dimension. Using hashtags as words, we look for an axis representing the political alignment in the embedding vector space. We leverage a dataset of political tweets collected during the 2018 U.S. midterm elections[57]. The hashtags from the same tweet are grouped together as a single sentence and fed to the word2vec algorithm to obtain vector representations for the hashtags. We remove hashtags appearing less than five times in the dataset, which may be a rare misspelling or too uncommon to obtain a reliable signal. Note that common variations and misspellings of a hashtag are treated similarly to the original one. After filtering, we end up with 54,533 hashtag vectors. To define the political alignment axis, we choose #voteblue and #votered as two poles because they show clear alignment with U.S. liberal and conservative political orientations, respectively. The rest of the hashtags are then projected onto this axis, and the relative positions, scaled into the interval $[-1, 1]$, are used to measure the political alignment where negative/positive scores indicate Left/Right alignment.

The link-based approach considers links (URLs) commonly used to share news and other websites in social media posts, for the purpose of spreading information, expressing opinions, or flagging identity, especially around political matters. Many websites show clear political bias. If the political ideology of a set of social media users is known, one can use it in conjunction with knowledge of the websites they share and like on the platform to infer the political bias of the sources[58]. Therefore, the websites (domains) extracted from links provide us with another convenient proxy for the political alignment of tweets and users. To assess the political alignment of a website, we start with a dataset of 19 thousand ideologically diverse news sources, where each domain is assigned a score reflecting its political alignment in the liberal-conservative range $[-1, +1]$. These scores are obtained from the sharing activity of Twitter accounts associated with registered U.S. voters[59,60]. For each link found in the tweets, we matched the domain to the list to obtain a score. See Supplementary "Methods" for additional details.

We further aggregate the political alignment scores of the tweets, obtained using either hashtags or links, at the account level. We examine different types of political alignment for accounts, each measured on a daily basis. The political alignment to which a drifter is exposed, $s_h$, is computed by averaging the scores of 50 most recent tweets from its home timeline. We also evaluate the political alignment expressed by an account by averaging recent tweets they post. We measure this expressed alignment $s_u$ for each drifter using its most recent 20 tweets. We use $s_f$ to represent the political alignment expressed by the friends of each drifter, using their 500 collective tweets. In Supplementary "Methods", we detail how political alignment scores are calibrated so that a value of zero can be interpreted as aligned with the political center.

Since $s_f$ represents the political alignment expressed by the friends of a drifter and $s_h$ represents the alignment of the posts to which the drifter is exposed in its home timeline, the difference $s_h - s_f$ can be used to measure potential political bias in Twitter's news feed.

**Identification of low-credibility content**. In evaluating the credibility of the content to which drifters are exposed, we focus on the sources of shared links to circumvent the challenge of assessing the accuracy of individual news articles[47]. Annotating content credibility at the domain (website) level rather than the link level is an established approach in the literature[33,49,61–63].

A low-credibility source is one that exhibits extreme bias, propaganda, conspiratorial content, or fabricated news. We use a list of low-credibility sources compiled from several recent research papers. Specifically, we consider a source as low-credibility if it fulfills any one of the following criteria: (1) labeled as low-credibility by Shao et al.[33]; (2) labeled as "Black," "Red," or "Satire" by Grinberg et al.[49]; (3) labeled as "fake news" or "hyperpartisan" by Pennycook et al.[62]; or (4) labeled as "extreme Left," "extreme Right," or "fake news" by Bovet et al.[63]. This provides us with a list of 570 sources.

To measure the percentage of low-credibility links, we extracted the links from the home timelines of the drifters (expanding those that are shortened) and then cross-referenced them with the list of low-credibility sources.

**Echo chambers**. We wish to measure the density and transitivity of each drifter bot's ego network. Since reconstructing the full network of friends and followers of each bot is prohibitively time-consuming due to the Twitter API's rate limits, we approximated each bot's ego network by sampling 100 random neighbors from a list of the latest 200 friends and 200 followers returned by the Twitter API. We then checked each pair of sampled neighbors for friendship. We added an undirected edge if there was a follower/friend link in either direction, so that the sampled ego network is undirected and unweighted. Finally, we computed the density and transitivity of each ego network[64].

Since transitivity is correlated with density, we also normalized the transitivity by the average transitivity of 30 shuffled networks generated by a configuration model that preserves the degree sequence of the original ego network. We replace any self-loop and parallel edges, generated by the configuration model, with random edges.

**Reporting Summary**. Further information on research design is available in the Nature Research Reporting Summary linked to this article.

## Data availability
To ensure the reproducibility of the experiment in this study, we share our code that generates a database of Twitter content. We do not share the resulting raw data to comply with Twitter terms that prohibit sharing content obtained from the Twitter API with third parties. We do provide code to process the raw data into an intermediate data format that includes all the derived information necessary for analysis. This pre-processed data are also shared. User and tweet IDs are anonymized to protect subject privacy. This allows other researchers to reproduce our results and/or compare their results.

The data are available in a public repository at Github (github.com/IUNetSci/DrifterBot) and at Zenodo (https://doi.org/10.5281/zenodo.4750190)[65]. Source data are also provided with this paper.

## Code availability
All of the code used to run the experiment and produce the figures in this manuscript is available in a public repository at Github (github.com/IUNetSci/DrifterBot) and at

Zenodo (https://doi.org/10.5281/zenodo.4750190)[65]. The repository lists dependencies on external libraries, such as twurl, chatterbot, gensim, and the Botometer Pro API and Python client library.

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

## Acknowledgements
We are grateful to Paul Cheung for a conversation that inspired the drifter experiment, and to Eni Mustafaraj for helpful suggestions. This work was supported in part by Knight Foundation and Craig Newmark Philanthropies. Any opinions, findings, and conclusions or recommendations expressed in this material are those of the authors and do not necessarily reflect the views of the funding agencies.

## Author contributions
F.M. and W.C. designed the study. W.C. and D.P. implemented the bot activities and the data collection. K.C.Y. developed the hashtag embedding method to infer political alignment scores. All authors analyzed the data and wrote the report.

## Competing interests
The authors declare no competing interests.
