## [Peer Review File · Nature Communications]

Neutral Bots Probe Political Bias on Social MediaReviewers' comments:

Reviewer #1 (Remarks to the Author):

The paper "Neutral Bots Reveal Political Bias on Social Media" presents a really interesting question: once neutral bots (i.e. without a biased behaviour) are programmed, how do they evolve, due to their initial interactions? Interestingly enough their evolution has not a strong dependence on the initial following accounts: the interactions of the bots with their friends drift almost all them on righter positions.

I judge the paper worth of publication, after some explanation from the authors.

The first important issue is the following.

Actually, (I can be wrong!), the botometer score (b-score) is a proxy of the probability that the account considered may be a bot. In this sense, the low b-score accounts are those that are more likely to be humans. Among others, this behaviour is due to the propensity of users to have quite a strong activity. Nevertheless, many genuine accounts are hardly distinguishable from bots since they limit their activity, in good faith. In this sense, focusing on low b-scores, you focus your attention on particularly active users, which do not necessarily represent the high majority of the users. In this sense, the conclusions may be biased: particularly active neutral bots are drifted due to their interactions with other users, but in principle such behaviour could be limited to the extremely active accounts.

Other remarks are much less crucial:

- In the Introduction a deeper review of the literature can be performed, for instance, regarding Echo Chambers and radicalisation.
- At page 7, the authors consider the total co-occurrences of hashtags. The overlap can be of limited information, once their occurrences are not discounted: the overlap can appear with an extremely viral hashtag just due to the frequency of its appearance. Do the authors consider the possibility of discounting the information of the occurrences? How are results modified by considering them? Please add a comment.
- In the Discussions, what it seems to be missing is a possible explanation (or otherwise, a final comment) regarding the nature of the observed phenomenon. May the drift toward right positions of all (neutral) bots be due to either the extreme activity of right wing accounts or their strong presence or the platform feed or to all of them? Following the same idea at the beginning of the present review, do the authors consider the possibility that less active users would experience a (relative) lower drift?
- At page 11, the authors say that drifters were inactive from midnight to 7 a.m. Has this behaviour been recovered from the humans' behaviour as detected by the botometer? Does the results change in changing this window?
- At page 15, authors say that they removed all hashtags appearing less than 5 times. Do they consider the possibility of finding misspells in the hashtags? Some of them can be extremely common and can alter the statistics. Please add a comment.

- Finally, at page 22 and 23 the figures are too small to distinguish the trajectories of all bots and it is hard to distinguish how similar they are. In this sense, if trajectories are too close, it may be that the initial friendships are enough to limit the variability of the evolution. Please increase the resolution of the figure and add comments on this issue.

Best regards

Reviewer #2 (Remarks to the Author):

The paper reports on experiment that lasted over 5 months on Twitter. The goal was to understand contribution of social media platform to political bias of the content it helps spreading and creation of so-called echo chambers.

The method relies on analysis of the content and the local social network created around the bots (designated as drifters) introduced by the researchers into Twitter. I was very excited to read about the experiment design. A few groups of identical bots created by the authors were set to follow twitter users. Each group composed of three drifters would initially follow the same Twitter account. Five such groups followed five distinct accounts with different political alignment. Each of the bots would implement a fairly simple strategy to execute random actions (e.g. tweets, retweets, likes and replies). The drifters imitate human activity by mining their feed and responding to activity of the surrounding users.

The primary paper results are measurements of the political alignment of the content consumed and produced by the bots, credibility of the content and the properties of the networks evolved around the bots.

The question raised in the paper is very current and of great interest and importance. I really like the use of bots to interact with Twitter platform.

I have two primary concerns about the paper.

First, the scale of the experiment is too small to support deep, generalizable conclusions.

The paper maps findings in each group of drifters onto the corresponding political alignment. Yet, five groups are initialized by following 5 (five!) user accounts that range from left to right of the political spectrum. I didn't find the details of how these users were selected (did I miss that?), but I find it difficult to generalize observations from such a small and localized sets of measurements onto the entire political spectrum. What 'left' and 'right' represent? There are so many different flavors. There is such a huge variance withing each political belief! I fear that the small scale of the experiment cannot support any generalization and there is no way to demonstrate whether the results apply to hundreds of millions of Twitter users.

Second, sadly, a flaw in the design of the experiment limits its value. The drifters introduced into the network by following different users are not identical (as claimed in the paper). Each drifter mines its own local information sources. In other words, the treatment they generate is biased in the same way as their network environment. The coupling between the local network and treatment make it impossible to separate the platform effects from naturally occurring biases (which is the whole point of conducting the experiment) or gaining insights into any of the processes causing such bias. True, one could use such bots to see how the social network and the content evolve over time. Yet, the bots are coupled to their environment and it's unclear whether the difference in the network properties (e.g. number of the followers reported in Fig 1 or the bots score reported in Fig 2) arise from the platform bias, the content bias (e.g. a larger share of fake content could have resulted from a larger number of bot followers, or the other way around), or any other reason. Frankly, I don't see significant benefits from introduction of homophilous bots and why the conclusions differ from tracking users that are naturally embedded in the network. On the contrary, authentic users would exhibit natural behavior (the authors acknowledge that Twitter users could

recognize drifters as bots and treat them accordingly). One could also increase the scale of the analysts by monitoring many such users.

Let me demonstrate this point with one of the findings. It is reported that more (politically) extreme drifters acquire followers at a faster rate than less extreme drifters leaning to the same side (Figure 1 and page 4). To measure this, one could have compared evolution of new accounts following more and less extreme twitter users. I'm not sure how introduction of the bots served the authors here. It's still unclear whether the effect is induced by the platform, by the behavior of the Twitter users with more and less extreme views (perhaps, extreme-leaning users are more engaged Twitter or create more social links, in which case the observed is not a platform bias; or it could be an implication of the Twitter recommender system, in which case it is) or some other reason. Similar arguments can be made for all other findings.

I'd consider refining experiment design to address these deficiencies. For instance, all five sets of drifters could produce identical treatment (e.g. content drawn from the same feed). One could then track whether the populations exposed to the same content respond differently or choose to spread politically flavored subset of that content. What kind of users would choose to dissipate less reliable content? Obviously, this is only one of the many possible approaches. My point is that to draw causal conclusions the treatment must not be coupled with response to the treatment.

I'd suggest revising the research question, which is currently too general (and isn't really answered in the paper). The experiment and the analysis are neat, but their implications are not really clear. I believe that more precise research question and experiment designed to address the question directly would make this work so much better!

Minor issues.

Although, the language is typically clear, the paper is not easy to follow. In particular, it took me a while to understand the setup of the experiment, implementation of drifters, selection of the originally followed accounts or how their political alignment was computed. All these details are provided later in the paper, in the Methods Section, which sometimes sent me further to Supplementary Information. This arrangement forced frequent back and forth scrolling and left a feeling of solving a quest rather than continuous reading. I think, that giving few very basic details earlier in the paper would improve its flow and make it much easier to follow.

From the figures in the supplementary, it seems that some bots were deactivated before the experiment end in December. Why?

It would be interesting to see how different metrics are correlated. For example, what is the cross-correlation between the user timeline and her home timeline?

The paper could benefit from additional descriptive statistics on the bots, including statistics about their friends, how many tweets included links, how many hashtags, etc.

Dear Reviewers,

Thank you for giving us an opportunity to submit a revised manuscript. We read the reviews carefully and appreciate the feedback. We feel that the revisions not only address each of the referee concerns, but also significantly strengthen the manuscript, so we are grateful. Below we respond to all reviewer comments and describe the manuscript revisions. Our responses are in blue.

In addition to the revisions discussed below, we also made another important improvement. We recently became aware of a new dataset of political scores associated with news sources (Robertson et al., 2018). While the methodology for this data is similar to the one used for the dataset we had used initially (Bakshy et al., 2015), the new data is more recent, includes more domains (19 thousand vs 500), and is based on Twitter accounts associated with registered voters in the U.S. Comparing the scores from the two datasets for the domains that are included in both, we found a strong consistency (Spearman correlation 0.95). For a few websites, the scores differ significantly across the two datasets. As an example, yournewswire.com went from a center-left score of -0.26 in the 2015 dataset to a far-right score of +0.7 in the 2018 dataset. We concluded that the 2018 scores are not only more recent and less sparse, but also more reliable for our present analysis.

Based on these considerations, we repeated our link-based analyses using the new dataset (Figs. 3, 5, S1, S2, S3, and Table S2). The results are more clear and consistent with the hashtag-based analyses. We therefore present these new results in the manuscript. We believe the paper is much improved as a result of the cleaner data and consistent conclusions.

Reviewer #1

The paper "Neutral Bots Reveal Political Bias on Social Media" presents a really interesting question: once neutral bots (i.e. without a biased behaviour) are programmed, how do they evolve, due to their initial interactions? Interestingly enough their evolution has not a strong dependence on the initial following accounts: the interactions of the bots with their friends drift almost all them on righter positions.

I judge the paper worth of publication, after some explanation from the authors.

We would like to thank the referee for their careful review, comments, and helpful suggestions!

The first important issue is the following.

Actually, (I can be wrong!), the botomer score (b-score) is a proxy of the probability that the account considered may be a bot. In this sense, the low b-score accounts are those that are more likely to be humans. Among others, this behaviour is due to the propensity of users to have quite a strong activity. Nevertheless, many genuine accounts are hardly distinguishable from bots since they limit their activity, in good faith. In this sense, focusing on low b-scores, you

focus your attention on particularly active users, which do not necessarily represent the high majority of the users. In this sense, the conclusions may be biased: particularly active neutral bots are drifted due to their interactions with other users, but in principle such behaviour could be limited to the extremely active accounts.

If we understand correctly, the referee is concerned that the drifter bots in our experiment tend to be particularly active in order to “look human” (low bot score), which could bias the results because they would not represent the majority of Twitter users. However, accounts with higher activity actually do *not* tend to have lower bot scores. We reproduce here a figure from a previous paper (Shao et al, Nature Communications 2018, doi.org/10.1038/s41467-018-06930-7) showing no correlation between activity and bot score. In other words, there are automated accounts with high and low activity.

More importantly, in our experiment, we do not focus on accounts that are particularly active. The drifter bots have typical activity levels for active Twitter accounts. Consider this plot comparing the distributions of activity (count of tweets and likes) by the drifters vs. a sample of active Twitter accounts. The means (triangles), medians (lines), and 50/95% confidence intervals (boxes and whiskers, respectively) of the two distributions are comparable.

Finally, if the referee is concerned about the bot scores or activity levels of friends and followers of the drifters (rather than the drifter accounts themselves), we point out that the experiment does not have any bias based on the bot scores of the accounts with which the drifters interact -- the accounts they retweet, like, follow, etc. The behavioral model is completely blind to any knowledge of characteristics of the accounts with which the drifters interact.

We revised the manuscript to make it clear that the drifter bots do not have unusual activity levels.

Other remarks are much less crucial:

- In the Introduction a deeper review of the literature can be performed, for instance, regarding Echo Chambers and radicalisation.

We added several references to literature on echo chambers and radicalization.

- At page 7, the authors consider the total co-occurrences of hashtags. The overlap can be of limited information, once their occurrences are not discounted: the overlap can appear with an extremely viral hashtag just due to the frequency of its appearance. Do the authors consider the possibility of discounting the information of the occurrences? How are results modified by considering them? Please add a comment.

We thank the reviewer for this comment, which made us notice that the summary description in the main text was inaccurate and led to confusion. The valence scores of the hashtags are not directly calculated from the co-occurrences. As detailed in the Methods section (second paragraph of the subsection “Political Alignment Metrics and Algorithmic Bias”), we apply the word2vec algorithm, which utilizes hashtag co-occurrence information to generate continuous vector representations of the hashtags. Such representations do take frequency information into account when encoding the semantic relationships between the hashtags; in other words, viral hashtags that co-occur with political hashtags do not distort the scores. The valence scores of the hashtags are then inferred from the relative position in the embedding space. Since the word2vec embedding algorithm already accounts for hashtag frequencies, no further frequency discounting is necessary.

We have revised the main text to clarify the use of embedding and avoid confusion.

- In the Discussions, what it seems to be missing is a possible explanation (or otherwise, a final comment) regarding the nature of the observed phenomenon. May the drift toward right positions of all (neutral) bots be due to either the extreme activity of right wing accounts or their strong presence on the platform feed or to all of them? Following the same idea at the beginning of the present review, do the authors consider the possibility that less active users would experience a (relative) lower drift?

We expanded the Discussion section drawing a more clear distinction between Twitter’s alleged algorithmic political bias (which our experiment rules out) and the strong bias that we do observe: the drifters are drawn to the conservative side, where they tend to receive more followers, find themselves in dense communities, follow more automated accounts, and are exposed to more low-credibility content. Since these phenomena cannot be attributed to bias in the platform’s algorithmic curation of the news feed, which appears to be neutral, they must be attributed to the use (and abuse) of the platform, and possibly the policies that govern this use. As noted by the reviewer, this means that the “right wing” of Twitter is more active. This leads to our concluding paragraph that emphasizes how neutral algorithms do not necessarily yield neutral outcomes.

- At page 11, the authors say that drifters were inactive from midnight to 7 a.m. Has this behaviour been recovered from the humans' behaviour as detected by the botometer? Does the results change in changing this window?

We based this ingredient of the model on the observation that people are inactive while they sleep. A 7-hour period of inactivity is consistent with social media data as reported by Barbosa

et al. (2018). We added a citation of this reference in the manuscript. We did not experiment with different sleep time windows, as this would be outside the scope of our analysis.

- At page 15, authors say that they removed all hashtags appearing less than 5 times. Do they consider the possibility of finding misspells in the hashtags? Some of them can be extremely common and can alter the statistics. Please add a comment.

Indeed, one of the reasons for removing hashtags that appear less than 5 times is to exclude rare misspellings. Our method would consider a common misspelling of a hashtag as semantically similar to the correctly-spelled hashtags. They would occupy nearby positions in the embedding space and be assigned similar scores. In fact, the word embedding technique has been used for spelling correction (Fivez et al. 2017). We added a comment in the paper to reflect this point.

Fivez P, Suster S, Daelemans W. Unsupervised context-sensitive spelling correction of clinical free-text with word and character n-gram embeddings. In Proc. BioNLP, pp. 143-148, 2017

- Finally, at page 22 and 23 the figures are too small to distinguish the trajectories of all bots and it is hard to distinguish how similar they are. In this sense, if trajectories are too close, it may be that the initial friendships are enough to limit the variability of the evolution. Please increase the resolution of the figure and add comments on this issue.

The journal will include high-resolution figures. In addition, full-resolution PDF (vector) images are available in the data repository. We added a comment about this in the supplementary text. The initial starting point for drifters in the same group are close but not identical by experimental design (we start from the same initial friend and a different random sample of 10 of their friends and followers). We observe that the trajectories diverge in several examples, suggesting that the initial conditions do not limit the variability of the evolution. We added a comment about this in the text.

Reviewer #2

The paper reports on experiment that lasted over 5 months on Twitter. The goal was to understand contribution of social media platform to political bias of the content it helps spreading and creation of so-called echo chambers.

The method relies on analysis of the content and the local social network created around the bots (designated as drifters) introduced by the researchers into Twitter. I was very excited to read about the experiment design. A few groups of identical bots created by the authors were set to follow twitter users. Each group composed of three drifters would initially follow the same Twitter account. Five such groups followed five distinct accounts with different political alignment. Each of the bots would implement a fairly simple strategy to execute random actions

(e.g. tweets, retweets, likes and replies). The drifters imitate human activity by mining their feed and responding to activity of the surrounding users.

The primary paper results are measurements of the political alignment of the content consumed and produced by the bots, credibility of the content and the properties of the networks evolved around the bots.

The question raised in the paper is very current and of great interest and importance. I really like the use of bots to interact with Twitter platform.

We would like to thank the referee for their careful review, comments, and helpful suggestions!

I have two primary concerns about the paper.

First, the scale of the experiment is too small to support deep, generalizable conclusions. The paper maps findings in each group of drifters onto the corresponding political alignment. Yet, five groups are initialized by following 5 (five!) user accounts that range from left to right of the political spectrum. I didn't find the details of how these users were selected (did I miss that?), but I find it difficult to generalize observations from such a small and localized sets of measurements onto the entire political spectrum. What 'left' and 'right' represent? There are so many different flavors. There is such a huge variance within each political belief! I fear that the small scale of the experiment cannot support any generalization and there is no way to demonstrate whether the results apply to hundreds of millions of Twitter users.

The referee is certainly right to observe that the number of drifters in our experiment is small compared to the millions of active Twitter accounts. The reasons for the small number are both operational and ethical. It is difficult to create and manage a large number of completely autonomous accounts over a long period of time and collect the vast amount of data they generate, while complying with an ethical research protocol (approved by the university's IRB) and the platform's terms of service and guidelines. For example, each account must be verified with a valid and unique phone number and email address, and is occasionally challenged by captcha, two-factor authentication, and other manual verification steps. Although the drifters do not impersonate people, they interact with human subjects, therefore a larger number of drifters could have a negative impact on the spread of misinformation, reinforcement of echo chambers, and amplification of malicious accounts. We designed the experimental protocol to achieve a trade-off between minimizing such potentially negative impact and maximizing the statistical significance of our results.

The referee notes that we consider only five "treatments" (groups of drifter bots) associated with five distinct initial friends. These sources were selected as they span the full range of the U.S. political spectrum, from left (liberal) to right (conservative). While we agree with the referee that mapping the political spectrum to a one-dimensional variable is reductive, this is done routinely in the literature and allows us to decrease the complexity of the experimental analyses. The use of a 5-point scale is also common in the literature; it is hard to classify subjects at a much higher resolution. As explained in the Methods section of the manuscript, we started from five Twitter

accounts associated with popular and active U.S. news sources (The Nation, The Washington Post, USA Today, The Wall Street Journal, and Breitbart News) that span the U.S. liberal-conservative spectrum.

Finally, we only have three 3 drifters per treatment (group). As mentioned above, this was a tradeoff between minimizing the total number of drifters (15) and having a sufficient number to achieve statistically significant results when comparing different groups.

That said, it would certainly be desirable to expand the experiment with more groups, more initial friends per group, and more drifters per initial friends in the future. Setting ethical issues aside, this would require a long time and may be all but impossible due to current platform changes. Yet, despite the limited number of drifters in our experiment, we argue that our results are significant. They provide important insights about the health of our information ecosystems, and will help raise awareness about biases of online interactions that affect our democracy, and inform the heated debate about platform bias that we're witnessing today as the US election approaches. If we are allowed to make an analogy, imagine launching a probe into an unexplored planet's orbit. The probe would only explore a tiny fraction of space, but that would not make the data it provides about the planet's gravity any less meaningful.

We revised the manuscript to discuss the factors leading to the limited number of drifters in our experimental protocol and the choice of treatment.

Second, sadly, a flaw in the design of the experiment limits its value. The drifters introduced into the network by following different users are not identical (as claimed in the paper). Each drifter mines its own local information sources. In other words, the treatment they generate is biased in the same way as their network environment. The coupling between the local network and treatment make it impossible to separate the platform effects from naturally occurring biases (which is the whole point of conducting the experiment) or gaining insights into any of the processes causing such bias. True, one could use such bots to see how the social network and the content evolve over time. Yet, the bots are coupled to their environment and it's unclear whether the difference in the network properties (e.g. number of the followers reported in Fig 1 or the bots score reported in Fig 2) arise from the platform bias, the content bias (e.g. a larger share of fake content could have resulted from a larger number of bot followers, or the other way around), or any other reason. Frankly, I don't see significant benefits from introduction of homophilous bots and why the conclusions differ from tracking users that are naturally embedded in the network. On the contrary, authentic users would exhibit natural behavior (the authors acknowledge that Twitter users could recognize drifters as bots and treat them accordingly). One could also increase the scale of the analysts by monitoring many such users.

Let me demonstrate this point with one of the findings. It is reported that more (politically) extreme drifters acquire followers at a faster rate than less extreme drifters leaning to the same side (Figure 1 and page 4). To measure this, one could have compared evolution of new accounts following more and less extreme twitter users. I'm not sure how introduction of the bots served the authors here. It's still unclear whether the effect is induced by the platform, by the

behavior of the Twitter users with more and less extreme views (perhaps, extreme-leaning users are more engaged Twitter or create more social links, in which case the observed is not a platform bias; or it could be an implication of the Twitter recommender system, in which case it is) or some other reason. Similar arguments can be made for all other findings.

These comments suggest that our manuscript was not sufficiently clear in presenting the experimental design, research questions, and “causal conclusions”. This may have led the referee to suspect a flaw in the design and to be confused about result claims. We extensively revised both the Introduction and Discussion sections of the manuscript to add clarity about attribution. We outline a few key points here.

To consider the initial conditions (friends) of the drifters as treatment, we claim that the drifters are identical. What we should have better explained is that the stochastic *behavior* of the drifters is identical. The *actions* that results from this behavior, however, are dependent on the treatment, as correctly noted by the referee -- for example when a drifter retweets one of its friends. This is part of the design, as the goal is to see how the actions and the local environment of each drifter are affected by its initial condition.

We stress that with one exception (see next paragraph) we do not aim, nor claim, to “separate the platform effects from naturally occurring biases” -- the referee is correct that this is impossible due to the coupling between the local network and treatment. But this is *not* the point of our experiment. Rather, the point is to see how the coupling of initial condition and information ecosystem (which includes both platform algorithms and other users) affects outcomes. If we tracked human users, as the referee suggests, it would be impossible to separate these effects from the biases of those individual humans. The use of neutral bots, on the other hand, removes the need to control for any number of confounding factors that might affect the actions of human users, such as age, gender, race, ideology, education, socio-economic status, etc.

There is only one analysis in the paper where we do “draw causal conclusions” about platform bias, namely, the news feed (home timeline) algorithm (Fig. 5(c,f)). We can separate this effect from the treatment because we measure the *difference* between the alignment of the home timeline and that of the local network’s content, from which the home timeline tweets are selected. For all other findings (Figs. 1, 2, 3, 4, 5(a,b,d,e)), the observed patterns are *not* causally attributed to platform bias. For example, we acknowledge in the manuscript that the observed bias in exposure to misinformation may be related to exposure to bots and echo-chamber structure.

The reviewer refers to “homophilous bots”. We would like to stress that the behavioral model of the drifter bots is *not* homophilous, despite their own local networks. The model completely ignores content and any other characteristics of accounts. As an illustration, one of the “conservative” drifters at one point followed @CNN and many “liberal” accounts. Actions can still be homophilous because of the social media mechanisms (eg, one is unlikely to retweet someone they and their friends do not follow), but that is part of what we analyze.

I'd consider refining experiment design to address these deficiencies. For instance, all five sets of drifters could produce identical treatment (e.g. content drawn from the same feed). One could then track whether the populations exposed to the same content respond differently or choose to spread politically flavored subset of that content. What kind of users would choose to dissipate less reliable content? Obviously, this is only one of the many possible approaches. My point is that to draw causal conclusions the treatment must not be coupled with response to the treatment.

This is a very interesting idea, and we may follow the recommendation in a future study! We refer the reviewer to our previous response regarding our clarification of the causal conclusions of the experiment.

I'd suggest revising the research question, which is currently too general (and isn't really answered in the paper). The experiment and the analysis are neat, but their implications are not really clear. I believe that more precise research question and experiment designed to address the question directly would make this work so much better!

We thank the referee for this suggestion. We rewrote the Introduction section to clarify our research question, as follows: *(i) How are influence and exposure to inauthentic accounts, political echo chambers, and misinformation impacted by early actions on a social media platform? (ii) Can such differences be attributed to political bias in the platform's news feed algorithm?* The text is now more explicit (in both Introduction and Discussion sections) about how we explore these questions with our experimental design.

Minor issues.

Although, the language is typically clear, the paper is not easy to follow. In particular, it took me a while to understand the setup of the experiment, implementation of drifters, selection of the originally followed accounts or how their political alignment was computed. All these details are provided later in the paper, in the Methods Section, which sometimes sent me further to Supplementary Information. This arrangement forced frequent back and forth scrolling and left a feeling of solving a quest rather than continuous reading. I think, that giving few very basic details earlier in the paper would improve its flow and make it much easier to follow.

While the structure of the paper (with Methods at the end) is imposed by the journal, we revised the manuscript to add more high-level descriptions of the methods earlier, in the Results section.

From the figures in the supplementary, it seems that some bots were deactivated before the experiment end in December. Why?

We revised the plots to improve clarity. The missing data was due to two reasons. The first, as the referee guessed, is that two drifter accounts were temporarily suspended in mid November. We added an explanation of this in the plot captions. We stress that there was no violation of Twitter terms; rather, Twitter conducted routine verifications of the phone numbers and we

neglected to notice this because we were not actively monitoring the associated email addresses. The second reason is that the links generated by the drifters are sparse, so, as explained in the Methods, missing values are interpolated. We had neglected to fill the missing values in the final period of the plots, and we have now fixed this oversight. Furthermore, the use of the new URL score dataset (Robertson et al., 2018, discussed above) ameliorates the link data sparsity.

It would be interesting to see how different metrics are correlated. For example, what is the cross-correlation between the user timeline and her home timeline?

Following this suggestion, we performed a time-lagged cross-correlation analysis between user and home timelines of the drifters, using both hashtags and links to calculate political valence scores. The table below reports the lags between home and user timelines yielding the maximal cross-correlation. Cases where the correlation is not significant ($p > 0.01$) are dropped.

	Hashtags		Links	
	Lag (days)	r	Lag (days)	r
Left	4	0.51	4	0.43
	–	–	1	0.82
	–	–	6	0.50
C. Left	1	0.59	–	–
	1	0.65	–	–
	2	0.69	2	-0.42
Center	–	–	–	–
	–	–	4	0.73
	–	–	–	–
C. Right	1	0.44	1	0.89
	–	–	4	0.92
	1	0.41	2	0.68
Right	–	–	4	0.54
	–	–	–	–
	–	–	–	–

We observe that in many cases the cross-correlation is not significant, and the maximal cross correlations are obtained for different lags. In one case (highlighted) the correlations is negative; this case corresponds to the drifter with both liberal and conservative accounts among its

friends (cf. Fig 3). We do not feel that these results yield particularly useful insights warranting inclusion in the manuscript or Supplementary Information. However, we will be happy to add them if the reviewer finds them informative.

The paper could benefit from additional descriptive statistics on the bots, including statistics about their friends, how many tweets included links, how many hashtags, etc.

We added in Supplementary Information a new table reporting descriptive statistics of the bots near the end of the experiment: number of friends, number of followers, number of original tweets, number of retweets, number of likes, number of hashtags posted, number of links posted, and total number of actions (Table S3).

We also added a new plot reporting on the overlap between friends and followers of the drifters (Fig. S4). We observe a higher follow-back rate for partisan drifters, and especially conservative ones.

Reviewers' comments:

Reviewer #1 (Remarks to the Author):

I read the answer of the authors, and i am impressed by the work they have done, I am totally satisfied by their extremely detailed answer. I recommend publication of the paper in the present form

Reviewer #3 (Remarks to the Author):

I would like to clarify that in my capacity as a reviewer, I am acting as a proxy for the previous Reviewer 2, who was unable to act as a reviewer in this subsequent revision. As such, I have been asked to focus my consideration on the points that R2 raised and to evaluate the authors response.

There two most important concerns pointed out by Reviewer 2 are:

1. The experiment is of too small a scale to draw meaningful conclusions that are generalizable to the platform as a whole
2. The experimental design is flawed because drifters do not act identically

Regarding the first point, I agree that a larger scale experiment would be desirable but am sympathetic to the authors' practical and ethical barriers to increasing the scale. While a large-scale experiment of a similar nature might be possible (esp. if conducted by the platform itself), it seems less likely that this would be pass Human Subjects Research guidelines and obtain IRB approval (by a platform IRB body or, university IRB if the research team involved university researchers). While there may be a large variety of politically-aligned news sources (with which drifter accounts could be initialized), the ones the authors selected are quite popular. To show that this is the case, the authors might consider including some metrics about what proportion of real twitter users that are designated as (left, c. left, center , c. right, right) follow the news sources that represent their five treatment groups (e.g., by analyzing a dataset of twitter users labelled by these five alignments). One potential issue that relates to this concern is that prior research has indicated that Brietbart News is of low credibility. While the authors remove this designation from metrics that assess credibility in the outcomes they measure, there is still the concern that the authors findings about the right-leaning treatment group (particularly those about exposure to low-credibility links) actually reflect its low credibility nature rather than its partisan nature. In other words, would the findings still hold for a treatment group that was initialized to follow a right-leaning news account that is not of low credibility? Testing this directly would require an additional experiment, though I do not the magnitude of the concern warrants this. Instead, the authors may i) defend their choice by claiming that Breitbart is the best parallel to the politically-aligned news sources in the other treatment groups (with some evidence that supports this assertion) and ii) mention this as a potential limitation of their study.

Regarding the second point, I agree with the authors that there was likely some confusion on the part of R2 surrounding the definition of the treatment. In complex social network environments, it is natural to design treatments that depend upon local network environments and while more "homogenous" treatments (i.e., bots that all act the same regardless of their evolving network environment) may be easily implemented, it isn't clear

what we might learn from such treatments. R2's comment seems to be more interested in an inside-out design (measure the impact of a drifter bot's behavior on its peers) than the outside-in design that the authors have chosen (measure the evolving information to which a drifter bot with treatment X is exposed). One could argue that the behavior of a drifter bot is likely to differ from organic twitter users (which I believe is the case), but this seems to be precisely the point of defining the drifter bot this way – to measure the impact of a Twitter user's initial (biased) action of following partisan accounts on information exposure in the absence of any subsequent biased behavior of that user. I think the authors of the manuscript tacitly understand that real users would not perform the random actions of drifter bots, but would instead act selectively (likely with the same partisan bias as their initial choice). Perhaps the authors can add a sentence or two to convey this understanding to readers more emphatically. Ultimately, I believe the experiment is well designed.

In summary, I believe the authors have sufficiently address the concerns discussed above. I suggested some very minor changes/additions to the paper. I recommend this manuscript for publication.

Reviewer #1:

I read the answer of the authors, and i am impressed by the work they have done, I am totally satisfied by their extremely detailed answer. I recommend publication of the paper in the present form.

We would like to thank the reviewer for the helpful review and kind words!

Reviewer #3:

I would like to clarify that in my capacity as a reviewer, I am acting as a proxy for the previous Reviewer 2, who was unable to act as a reviewer in this subsequent revision. As such, I have been asked to focus my consideration on the points that R2 raised and to evaluate the author's response.

We appreciate the referee's generous service in stepping in for R2 and helping us improve the manuscript.

There two most important concerns pointed out by Reviewer 2 are:

1. The experiment is of too small a scale to draw meaningful conclusions that are generalizable to the platform as a whole
2. The experimental design is flawed because drifters do not act identically

Regarding the first point, I agree that a larger scale experiment would be desirable but am sympathetic to the authors' practical and ethical barriers to increasing the scale. While a large-scale experiment of a similar nature might be possible (esp. if conducted by the platform itself), it seems less likely that this would be pass Human Subjects Research guidelines and obtain IRB approval (by a platform IRB body or, university IRB if the research team involved university researchers). While there may be a large variety of politically-aligned news sources (with which drifter accounts could be initialized), the ones the authors selected are quite popular. To show that this is the case, the authors might consider including some metrics about what proportion of real twitter users that are designated as (left, c. left, center, c. right, right) follow the news sources that represent their five treatment groups (e.g., by analyzing a dataset of twitter users labelled by these five alignments).

That is a great suggestion. Accordingly, we analyzed the set of news sources categorized by AllSides.com (<https://www.allsides.com/media-bias/media-bias-ratings>) as having a left, center-left, center, center-right, or right bias. We performed two analyses. First, we report on the number of followers of each Twitter account corresponding to these sources.

Second, as suggested by the reviewer, we started from a large sample of Twitter accounts who tweeted on a particular day during the experiment. We excluded likely bots to focus on “real Twitter users.” We then classified a sample of these users, based on the links they shared, into the five categories (left, c. left, center, c. right, right). Finally, for each group, we counted the proportion that followed each of the news sources in the AllSides.com set. This shows that the sources we used as seeds are among the most followed by real users in each group.

Method:

- (1) We started with the raw Decahose (10%) tweets on 2019-08-01, which contains about 36,000,000 tweets from 13,825,527 unique accounts
- (2) We sampled 500,000 unique accounts; for each account, we also extract one tweet from them.
- (3) We calculated the bot score (using BotometerLite) for these 500,000 accounts using the tweet extracted and got the language from that tweet as their language
- (4) We removed non-English accounts and got 164,821 accounts
- (5) We removed the ones with bot scores larger than 0.5 and ended up with 151,570 accounts
- (6) We extracted a week of tweets by those 151,570 accounts from the Decahose (10% random sample of public tweets), during one week around 2019-08-01.
- (7) We used the links shared in those tweets to assign a political score to each of those accounts. We filtered out accounts for which we could not assign a score. That left 26304 accounts with a political score.
- (8) We grouped these 26304 accounts into 5 political bins using thresholds -1, -0.5, -0.1, +0.1, +0.5, +1, yielding groups of 712, 5280, 11422, 8237, 653 accounts respectively.
- (9) We examined the friends of these accounts and eliminated those who do not follow any of the 64 new sources from AllSides. 4884 accounts remained: 187, 1362, 1623, 1444, 268 in the five groups, respectively.
- (10) We measured the proportions of accounts in each group following each of the sources from AllSides.

The new Table S1 shows that the five sources we used as seeds are among the most popular overall as well as in each group. They are not necessarily *the* most popular. For example, in the left group, *The Nation* is not as popular as *HuffPost*, but was selected because its political alignment was further to the left (-0.7) compared to *HuffPost* (-0.3). We added both analyses as a new section along with Table S1 in the supplementary methods, and revised the main text pointing to this section to justify our selection.

One potential issue that relates to this concern is that prior research has indicated that Breitbart News is of low credibility. While the authors remove this designation from metrics that assess credibility in the outcomes they measure, there is still the concern that the authors findings about the right-leaning treatment group (particularly those about exposure to low-credibility links) actually reflect its low credibility nature rather than its partisan nature. In other words, would the findings still hold for a treatment group that was initialized to follow a right-leaning news account that is not of low credibility? Testing this directly would require an additional experiment, though I do not think the magnitude of the concern warrants this. Instead, the authors may i) defend their choice by claiming that Breitbart is the best parallel to the politically-aligned news sources in the other treatment groups (with some evidence that supports this assertion) and ii) mention this as a potential limitation of their study.

We concur and appreciate the suggestion. We added a paragraph in the Discussion section, in which we justify the choice of Breitbart as one of the most popular conservative news sources on Twitter (see also analyses mentioned above). An important factor to consider is that, based on literature showing a correlation between conservative leanings and vulnerability to misinformation (e.g. Grinberg et al 2019), experiments that attempt to factor out this correlation may yield a less-representative sample of politically active accounts, therefore potentially biased results. We also mention the possible limitations stemming from our choice and the need to corroborate our findings with future experiments using alternative seed sources.

Regarding the second point, I agree with the authors that there was likely some confusion on the part of R2 surrounding the definition of the treatment. In complex social network environments, it is natural to design treatments that depend upon local network environments and while more “homogenous” treatments (i.e., bots that all act the same regardless of their evolving network environment) may be easily implemented, it isn’t clear what we might learn from such treatments. R2’s comment seems to be more interested in an inside-out design (measure the impact of a drifter bot’s behavior on its peers) than the outside-in design that the authors have chosen (measure the evolving information to which a drifter bot with treatment X is exposed). One could argue that the behavior of a drifter bot is likely to differ from organic twitter users (which I believe is the case), but this seems to be precisely the point of defining the drifter bot this way – to measure the impact of a Twitter user’s initial (biased) action of following partisan accounts on information exposure in the absence of any subsequent biased behavior of that user. I think the authors of the manuscript tacitly understand that real users would not perform the random actions of drifter bots, but would instead act selectively (likely with the same partisan bias as their initial choice). Perhaps the authors can add a sentence or two to convey this understanding to readers more emphatically. Ultimately, I believe the experiment is well designed.

Thank you very much for your careful reading of our experimental design. The suggestion to emphasize neutrality over realism of the bot behavior is excellent. We added sentences in both Results and Methods sections acknowledging that the behavior of drifter bots is intended to be neutral, not necessarily realistic.

In summary, I believe the authors have sufficiently address the concerns discussed above. I suggested some very minor changes/additions to the paper. I recommend this manuscript for publication.

Thank you again for the extremely helpful feedback. We hope that the current revision has adequately incorporated the suggested changes and additions.

Reviewers' comments:

Reviewer #1 (Remarks to the Author):

I consider this paper worth of publication in the present form

Reviewer #3 (Remarks to the Author):

In this latest revision, the authors have addressed my prior concerns. They performed supplementary analysis to show that the sources they chose represent popular ones shared by twitter users (labelled) across different parts of the political partisanship spectrum. Two additional questions were raised by the editorial team in the course of the review and brought to my attention:

1) Whether the scale of the experiment was sufficient, to support the claims the authors have made.

2) Whether it is appropriate to use the term “credibility” as an attribute of sources (rather than as an attribute of the tweet).

Regarding (1), the authors findings indicate: a) that drifter bots that are initialized to the left or right tend to end up in “echo chambers” (clustered regions of the network with partisan affiliation) significantly more than bots initialized to more centered sources (Figure 3); b) that exposure to low credibility sources is significantly higher for bots initialized to right and center-right sources (Figure 4). This suggests that the scale is sufficient to uncover these differences. Of course, a larger scale experiment might yield results which are more robust to the chosen sources and tighter confidence in the results – but these are marginal improvements. In my opinion, the authors have sufficiently described the limitations of their work and are not overclaiming their results.

Regarding (2), I feel strongly that it is appropriate to define credibility as an attribute of the sources. A great deal of recent research has focused on the spread of misinformation and its relationship to partisanship. As many fact-checking organizations indicate, even “mainstream” news sources publish articles that vary in credibility and the extent to which they contain misinformation or are misleading. Efforts in the academic community of researchers studying misinformation have converged to shared databases that associate both credibility and partisan “skew” or “bias” with popular news sharing domains such as those the authors employed in this study.

It is also true that credibility can be further associated with a tweet itself. That is, a Twitter user may provide their own commentary, opinions or “facts” when they share a link. It is difficult to measure credibility in this sense, as it not easily accomplished in an automated approach (with natural language processing techniques). To do so robustly may require manual coding that is tantamount to fact checking, which may require extensive training of manual coders. But, from one perspective, users that share low credibility links – even if they refute the content of the links in their tweet – are still effectively promoting links to their followers. The authors are careful to define their measures as “exposure” to low credibility links. In that sense, I think it is acceptable that they do not delve further into the content.

Overall, I am satisfied with the current revision of the paper and recommend its publications.

Reviewer #4 (Remarks to the Author):

I've been asked to comment on the authors' rebuttal to previous rounds of review. My thoughts below:

1) Study design: I appreciate the discussion regarding scale, including the authors' response that a larger study might raise additional ethical concerns. A study with more drifters, using a greater number of seed accounts, would certainly better address questions about (a) the extent to which the results are idiosyncratic (i.e., driven by variability due to the stochasticity of the drifters' behavior), and (b) the choice of starter accounts. Since the behavior model is specified ex ante, the more pressing concern is (b). And here, there is a potentially serious issue related to the discussion in the second round of review: the ideological slant of starter accounts could be confounded with popularity.

Specifically, Table S1 shows that Breitbart is substantially more popular within its ideological quintile (followed by 61% in the sample) than the other seed accounts (followed by 14-43%). This is nontrivial because key results involve feedback dynamics driven by engagement and network structure (e.g., modularity). At the very least, this would affect the interpretation of results on influence and echo chambers. Now, it's possible given the work of Faris, Benkler et al. that asymmetries in network structure between left and right are relevant differences, so Breitbart's greater popularity reflects right-wing density rather than a worrisome confound. But this is a big assumption to make, and I don't see this consideration discussed in the text. From a research design perspective, I don't understand why the authors didn't simply choose starter accounts so that they were roughly comparable on the "followers within group" metric (say ~20%).

2) Confounding ideology and credibility: I'm less worried about Breitbart being unusually low-quality, since the research on right-wing sites being embedded within a larger ecosystem in which misinformation easily spreads is pretty clear (again, see Faris et al.). But choosing something like National Review as a starter account for the rightmost quintile would certainly help address this concern along with the popularity confound.

3) One point not raised in the original reviews: It's really not clear how to interpret the bot score results. The authors have designed a series of bots and set them loose to test for platform bias; what do we learn by analyzing the bot scores of those bots' own friends and followers? (What are the bot scores of the drifters?)

Thank you for giving us an opportunity to submit a revised manuscript. We read the reviews carefully and appreciate the feedback. As asked, we include here a point-by-point rebuttal. Our responses are in blue font. We hope that this will address the remaining concerns. We also include a version of the manuscript with all revisions highlighted.

Reviewer #1

I consider this paper worth of publication in the present form

We thank the referee again for their review of our manuscript.

Reviewer #3

In this latest revision, the authors have addressed my prior concerns. They performed supplementary analysis to show that the sources they chose represent popular ones shared by twitter users (labelled) across different parts of the political partisanship spectrum. Two additional questions were raised by the editorial team in the course of the review and brought to my attention:

- 1) Whether the scale of the experiment was sufficient, to support the claims the authors have made.
- 2) Whether it is appropriate to use the term “credibility” as an attribute of sources (rather than as an attribute of the tweet).

Regarding (1), the authors findings indicate: a) that drifter bots that are initialized to the left or right tend to end up in “echo chambers” (clustered regions of the network with partisan affiliation) significantly more than bots initialized to more centered sources (Figure 3); b) that exposure to low credibility sources is significantly higher for bots initialized to right and center-right sources (Figure 4). This suggests that the scale is sufficient to uncover these differences. Of course, a larger scale experiment might yield results which are more robust to the chosen sources and tighter confidence in the results – but these are marginal improvements. In my opinion, the authors have sufficiently described the limitations of their work and are not overclaiming their results.

Regarding (2), I feel strongly that it is appropriate to define credibility as an attribute of the sources. A great deal of recent research has focused on the spread of misinformation and its relationship to partisanship. As many fact-checking organizations indicate, even “mainstream” news sources publish articles that vary in credibility and the extent to which they contain misinformation or are misleading. Efforts in the academic community of researchers studying misinformation have converged to shared databases that associate both credibility and partisan

“skew” or “bias” with popular news sharing domains such as those the authors employed in this study.

It is also true that credibility can be further associated with a tweet itself. That is, a Twitter user may provide their own commentary, opinions or “facts” when they share a link. It is difficult to measure credibility in this sense, as it is not easily accomplished in an automated approach (with natural language processing techniques). To do so robustly may require manual coding that is tantamount to fact checking, which may require extensive training of manual coders. But, from one perspective, users that share low credibility links – even if they refute the content of the links in their tweet – are still effectively promoting links to their followers. The authors are careful to define their measures as “exposure” to low credibility links. In that sense, I think it is acceptable that they do not delve further into the content.

Overall, I am satisfied with the current revision of the paper and recommend its publications.

We sincerely thank the reviewer for their careful evaluation of our revised manuscript.

Reviewer #4

I've been asked to comment on the authors' rebuttal to previous rounds of review. My thoughts below:

1) Study design: I appreciate the discussion regarding scale, including the authors' response that a larger study might raise additional ethical concerns. A study with more drifters, using a greater number of seed accounts, would certainly better address questions about (a) the extent to which the results are idiosyncratic (i.e., driven by variability due to the stochasticity of the drifters' behavior), and (b) the choice of starter accounts. Since the behavior model is specified ex ante, the more pressing concern is (b). And here, there is a potentially serious issue related to the discussion in the second round of review: the ideological slant of starter accounts could be confounded with popularity.

Specifically, Table S1 shows that Breitbart is substantially more popular within its ideological quintile (followed by 61% in the sample) than the other seed accounts (followed by 14-43%). This is nontrivial because key results involve feedback dynamics driven by engagement and network structure (e.g., modularity). At the very least, this would affect the interpretation of results on influence and echo chambers. Now, it's possible given the work of Faris, Benkler et al. that asymmetries in network structure between left and right are relevant differences, so Breitbart's greater popularity reflects right-wing density rather than a worrisome confound. But this is a big assumption to make, and I don't see this consideration discussed in the text. From a research design perspective, I don't understand why the authors didn't simply choose starter accounts so that they were roughly comparable on the "followers within group" metric (say ~20%).

We acknowledge that we selected the seed sources to be popular, not “equally popular” among sources having similar political slant. Doing this would require running a completely new experiment. However, given changes in the platform, this would mean losing important findings from this experiment that remain valid. So instead we address that there are potential confounding factors in the discussion.

We explore the possibility of “popularity within ideological quantile” being a confounding factor by examining its relationship with the two dependent variables that the reviewer suggests might be most affected: (a) drifter influence (number of followers, Fig 1) and (b) drifter echo chambers (normalized transitivity, Fig 3). We show scatter plots and correlation values below:

The low and/or statistically insignificant correlations suggest that popularity within ideological quantile is a weak confounding factor at best. This evidence is consistent with the interpretation of our observations as stemming more from differences in the characteristics of the network, which are correlated with partisanship (in particular, a higher density of the right-wing network), than from the popularity of the seeds. This is also consistent with other results that have shown that echo chambers and partisanship are correlated and that there is a stronger echo chamber on the right (Benkler et al. 2018; Nikolov et al. 2021).

We added the following paragraph in the Discussion session to acknowledge the potential confounding factors of the seed sources:

“There are several possible confounding factors stemming from our choice of news source accounts used as initial friends of the \drifters{}: their influence as measured by the number of followers, their popularity among users with similar ideology, their activity in terms of tweets, and so on. For example, \mention{FoxNews} was popular but inactive at the time of the experiment. Furthermore, these quantities vary greatly both within and across ideological groups (see \SI{}). We selected popular news sources across the political spectrum, but it is impossible to control for all of these factors with a limited number of \drifters{}. We checked whether the popularity of an initial friend among sources with similar political bias could be a confounding factor in our analyses. We did not find a significant correlation between this measure and outcomes that might be affected, namely, \drifter{} influence and ego network transitivity. This suggests that our results can be attributed to differences in the characteristics of the network, which are

correlated with the political alignment of the initial friends, rather than to the popularity of the initial friends. This interpretation is consistent with a correlation between echo chambers and partisanship, especially on the political right, as reported in the literature~\cite{benkler2018network, Nikolov2020partisanship}.”

2) Confounding ideology and credibility: I'm less worried about Breitbart being unusually low-quality, since the research on right-wing sites being embedded within a larger ecosystem in which misinformation easily spreads is pretty clear (again, see Faris et al.). But choosing something like National Review as a starter account for the rightmost quintile would certainly help address this concern along with the popularity confound.

As in response to the previous point, using a different seed source for the conservative group of drifter bots would require running a completely new experiment. Given changes in the platform, this would mean losing important findings from this experiment that remain valid. However, it is important to underscore that we addressed this confounding factor already: we did **not** consider the Breitbart website as a low-credibility source in our analysis. Therefore, we believe that the results in Fig. 4 cannot be attributed to confounding ideology and credibility: the low-credibility links refer to sources other than Breitbart.

The discussion mentions the possible limitation stemming from our choice of Breitbart and the need to corroborate our findings with future experiments using alternative seed sources:

“Although \textit{Breitbart News} was not labeled as a low-credibility source in our analysis, our finding might still be biased in reflecting this source's low credibility in addition to its partisan nature. However, \mention{BreitbartNews} is one of the most popular conservative news sources on Twitter (see \SI{}). While further experiments may corroborate our findings using alternative sources as initial friends, attempting to factor out the correlation between conservative leanings and vulnerability to misinformation~\cite{grinberg2019fake, Nikolov2020partisanship} may yield a less-representative sample of politically active accounts.”

3) One point not raised in the original reviews: It's really not clear how to interpret the bot score results. The authors have designed a series of bots and set them loose to test for platform bias; what do we learn by analyzing the bot scores of those bots' own friends and followers? (What are the bot scores of the drifters?)

The figure shows a wide range of bot scores among the 15 drifters, computed on 20 October 2019 (around the middle of the experiment).

As we write in the manuscript, the literature has provided ample evidence

that social bots were used to influence online discussions about recent U.S. elections in 2016 and 2018. Therefore we aim to quantify exposure to likely automated Twitter accounts by the drifters, and its dependence on the slant of the seed sources — not because the drifters are themselves bots, but because they gauge the likelihood that any user interested in U.S. news might have similar exposure. It is not surprising to find likely bots *following* the drifters. However, finding likely bots among the *friends* of the drifters is significant because it gauges the *vulnerability* of users to this kind of online manipulation. Our results (Fig. 2 in the manuscript) show that the vulnerability is higher among partisan accounts on both sides of the political spectrum. Accordingly, we revised the description of this result:

“Unsurprisingly, drifters have likely bots among their followers, without significant differences across the political spectrum. Focusing on the friends reveals a more serious potential vulnerability of social media users. We find that accounts followed by centrist, moderate, and partisan drifters are increasingly more bot-like, respectively. Among partisan groups, right-leaning drifters tend to follow significantly more likely bots than left-leaning ones.”

We also added the following sentence in the Discussion (at the end of the paragraph discussing the social bot results), to help the readers interpret these results:

“Nevertheless, our results are consistent with findings that partisanship on both sides of the political spectrum increases vulnerability to manipulation by social bots~\cite{Yan2020partisanbots}.”

Reviewers' comments:

Reviewer #4 (Remarks to the Author):

I thank the authors for their responses to my concerns and those of the other reviewers. But I want to push back on one of their points in the memo: the correlation between seed popularity within ideological quintile and number of followers is both strong and significant. Thus you can't simply wish away the possible confounding of ideology and popularity. The most important thing I'd ask the authors to do is to think through how such confounding could bias results in either direction, and to add this to the discussion. Specifically, how would these results from the abstract be affected?

"The interactions of conservative accounts are skewed toward the right, whereas liberal accounts are exposed to moderate content shifting their experience toward the political center. Partisan accounts, especially conservative ones, tend to receive more followers, follow more automated accounts, and find themselves in dense communities."

We include here a point-by-point rebuttal. Our responses are in blue font. We hope that this will address the remaining concern. We also include a version of the manuscript with all revisions highlighted.

Reviewer #4

I thank the authors for their responses to my concerns and those of the other reviewers. But I want to push back on one of their points in the memo: the correlation between seed popularity within ideological quintile and number of followers is both strong and significant. Thus you can't simply wish away the possible confounding of ideology and popularity. The most important thing I'd ask the authors to do is to think through how such confounding could bias results in either direction, and to add this to the discussion. Specifically, how would these results from the abstract be affected?

"The interactions of conservative accounts are skewed toward the right, whereas liberal accounts are exposed to moderate content shifting their experience toward the political center. Partisan accounts, especially conservative ones, tend to receive more followers, follow more automated accounts, and find themselves in dense communities."

We thank the reviewer for the suggestion. We expanded the analysis of the correlation between drifter influence and seed account popularity. We added new plots (Fig. S1 in Supplementary Analysis, also shown below), considering not only the popularity of the seeds within political groups, but also their overall popularity/influence. We find no significant correlation between the numbers of followers of drifters and seed accounts (Pearson's $r=0.05$, $p=0.85$). However, as discussed in our previous rebuttal, the drifter influence is correlated with the popularity of the seeds among active accounts with similar political alignment (Pearson's $r=0.52$, $p=0.05$).

We added the following paragraph in the Results section:

The differences in influence among \drifters\ could be affected not only by the political alignment, but also by other characteristics of their initial friends. To disentangle these factors, we measured the correlation between the number of \drifter\ followers and two features of their initial friends: their overall influence and their popularity among other politically aligned accounts. While \drifter\ influence is not affected by the overall influence of the initial friends, it is correlated with their popularity among politically aligned accounts (see Supplementary Information). This is consistent with evidence that social tie formation is associated with shared partisanship~\cite{Moslehe2022761118}, which we further explore next.

In the revised manuscript, the confounding factor and its implication in the interpretation of the results are also discussed in the Discussion section. In summary, the effect of the political alignment of the initial friends is mediated by political homophily, as suggested by the reviewer's prior comments. Because of the connection between influence and homophily (shared partisanship), we reordered the results subsections so that the echo chamber analysis follows the above subsection on influence.

We modified the paragraph about confounding factors in the Discussion section as follows:

We selected popular news sources across the political spectrum as initial friends of the \drifters\ . There are several possible confounding factors stemming from our choice of these accounts: their influence as measured by the number of followers, their popularity among users with similar ideology, their activity in terms of tweets, and so on. For example, @FoxNews was popular but inactive at the time of the experiment. Furthermore, these quantities vary greatly both within and across ideological groups (see Supplementary Information). While it is impossible to control for all of these factors with a limited number of \drifters\ , we checked for a few possible confounding factors. We did not find a significant correlation between initial friend influence or popularity measures and \drifter\ ego network transitivity. We also found that the influence of an initial friend is not correlated with \drifter\ influence. However, the popularity of an initial friend among sources with similar political bias is a confounding factor for \drifter\ influence. Online influence is therefore affected by the echo-chamber characteristics of the social network, which are correlated with partisanship, especially on the political right~\cite{benkler2018network, Nikolov2020partisanship}. In summary, accounts following more partisan news sources receive more politically aligned followers, becoming embedded in denser echo chambers and gaining influence within those partisan communities.

We hope that these revisions will address the remaining concern of the reviewer. We feel that rather than weakening our results, they strengthen them by providing an interpretation for the finding of greater influence by drifters following partisan news sources, namely the likelihood of receiving more followers within partisan echo chambers. Therefore we are very grateful to the reviewer!

Reviewers' comments:

Reviewer #4 (Remarks to the Author):

It's unclear whether the t-tests reported in the main text are one-sided or two-sided. If they are not two-sided, they need to be redone as there are not strong prior expectations about the sign. Please also specify that the tests are two-sided in the text.

Dear Reviewer,

Thank you for your additional review of our revised manuscript. Below is our response, in blue font. We also include versions of the manuscript and supplementary information with revisions highlighted. Note that several additional revisions have been made in response to editorial requests.

Reviewer #4:

Remarks to the Author:

It's unclear whether the t-tests reported in the main text are one-sided or two-sided. If they are not two-sided, they need to be redone as there are not strong prior expectations about the sign. Please also specify that the tests are two-sided in the text.

Indeed, we had initially applied one-sided t-tests, but we understand that two-sided tests are more appropriate and we appreciate the reviewer's suggestion on this point. We have changed all t-tests to be two-sided, clarified this both in the main text and in the Supplementary Information, and updated the p-values accordingly.

Although a few of the t-tests presented in the earlier version of the manuscript are no longer significant, most claims in the manuscript are not affected and are supported by the tests that remain significant. Below we list our claims (from the abstract) and the associated tests (when applicable), including any changes.

1. "We find no strong or consistent evidence of political bias in the news feed."

This claim is supported by visual inspection of Fig. 5(c,f) and the t-tests in Supplementary Table 3. We changed these t-tests to be two-sided. Although the p-values change as can be seen in the updated supplementary table, the results do not change and therefore we did not make any changes to the main manuscript.

2. "Despite this, the news and information to which U.S. Twitter users are exposed depend strongly on the political leaning of their early connections. The interactions of conservative accounts are skewed toward the right, whereas liberal accounts are exposed to moderate content shifting their experience toward the political center."

These claims are supported by visual inspection of the trends in Fig. 5(a,b,d,e). The time series plots indicate standard errors. The results are unchanged.

3. "Partisan accounts, especially conservative ones, tend to receive more followers"

This claim is based on visual inspection of Fig. 1 and statistical tests. The two-sided t-tests supporting this claim remain significant, even after applying the Bonferroni correction accounting for multiple comparisons (as requested by the editor). In the supplementary notes, for

robustness, we report on two additional methods to compare the follower growth rates. The second method is based on a regression analysis and unchanged (reported differences are still significant). The third method combines regression analysis to estimate the growth rates and t-tests to compare these estimates. When using two-sided tests and Bonferroni corrections, this method yields consistent but not statistically significant differences, due to the low number of degrees of freedom.

4. “[Partisan accounts, especially conservative ones], find themselves in denser communities”

This claim was supported by visual inspection of Fig. 2 and statistical tests. After applying two-sided t-tests and Bonferroni corrections, the differences in density, transitivity, and normalized transitivity between Right and Center drifters all remain significant. The differences between Left and Center drifters, and between Left and Right drifters, are no longer significant after the Bonferroni correction. We reworded the claims in the abstract and discussion section accordingly, only referring to the Right vs. Center groups in regards to echo chambers.

5. “[Partisan accounts, especially conservative ones,] follow more automated accounts.

This claim is supported by visual inspection of Fig. 3 and statistical tests. After applying two-sided tests and Bonferroni correction, the only differences that are no longer significant are between Left and C. Left drifters, and between Right and Left drifters. Since the differences between Right and Center drifters and between Left and Center drifters remain significant, we did not change the claim in the manuscript.

6. “Conservative accounts are also exposed to more low-credibility content.”

This claim is supported by visual inspection of Fig. 4 and statistical tests. The only difference that is not statistically significant after applying the Bonferroni correction is between Right and C. Right drifters. Therefore we consider the claim valid and did not change the manuscript.